# Association with PD-L1 Expression and Clinicopathological Features in 1000 Lung Cancers: A Large Single-Institution Study of Surgically Resected Lung Cancers with a High Prevalence of *EGFR* Mutation

**DOI:** 10.3390/ijms20194794

**Published:** 2019-09-26

**Authors:** Seung Eun Lee, Yu Jin Kim, Minjung Sung, Mi-Sook Lee, Joungho Han, Hong Kwan Kim, Yoon-La Choi

**Affiliations:** 1Department of Pathology, Konkuk University Medical Center, Konkuk University School of Medicine, Seoul 05030, Korea; 20150063@kuh.ac.kr; 2Laboratory of Cancer Genomics and Molecular Pathology, Samsung Medical Center, Sungkyunkwan University School of Medicine, Seoul 06351, Korea; bubble@skku.edu (Y.J.K.); mjsung0111@gmail.com (M.S.); goindol79@hanmail.net (M.-S.L.); 3Department of Pathology and Translational Genomics, Samsung Medical Center, Sungkyunkwan University School of Medicine, Seoul 06351, Korea; joungho.han@samsung.com; 4Department of Thoracic and Cardiovascular Surgery, Samsung Medical Center, Sungkyunkwan University School of Medicine, Seoul 06351, Korea; 5Department of Health Sciences and Technology, SAIHST, Sungkyunkwan University, Seoul 06351, Korea

**Keywords:** lung cancer, immunotherapy, PD-L1, *EGFR* mutation

## Abstract

Programmed cell death ligand 1 (PD-L1) expression is an important biomarker for predicting response to immunotherapy in clinical practice. Hence, identification and characterization of factors that predict high expression of PD-L1 in patients is critical. Various studies have reported the association of PD-L1 expression with driver genetic status in non-small cell cancer; however, the results have been conflicting and inconclusive. We analyzed the relationship between PD-L1 expression and clinicopathological factors including driver genetic alterations in 1000 resected lung cancers using a clinically validated PD-L1 immunohistochemical assay. PD-L1 expression was significantly higher in squamous cell carcinoma (SCC) compared to adenocarcinomas. PD-L1 expression in adenocarcinoma was associated with higher N-stage, solid histologic pattern, *EGFR* wild type, and *ALK* positive, but no significant association with the clinicopathological factors in SCC. *EGFR* mutant adenocarcinomas with distinctive clinicopathologic features, especially solid histologic pattern and higher stage showed higher PD-L1 expression. To the best of our knowledge, this study is the largest to evaluate the association between PD-L1 expression and clinicopathological and molecular features in lung cancer with a highly prevalent *EGFR* mutation. Therefore, our results are useful to guide the selection of lung cancer, even *EGFR*-mutated adenocarcinoma patients with PD-L1 expression, for further immunotherapy.

## 1. Introduction

The immune checkpoint inhibitors anti-programmed cell death 1 (PD-1)/programmed cell death ligand 1 (PD-L1) (anti–PD-1/PD-L1) are currently changing the approach of treatment to patients with non-small cell lung cancer (NSCLC). Over the last two years, the U.S. Food and Drug Administration (FDA) has granted approval to the anti–PD-1 inhibitors, nivolumab (OPDIVO) and pembrolizumab (KEYTRUDA) and the anti–PD-L1 inhibitor, atezolizumab (TECENTRIQ) for the treatment of patients with advanced NSCLC with progression during or after first-line therapy [1,2,3].

Identification of patients who might benefit from immune checkpoint inhibition in NSCLC is significant. PD-L1 expression is an important biomarker for the prediction of response to anti-PD1 and anti-PD-L1 immunotherapy. Therefore, the evaluation of PD-L1 expression in cancer is of critical concern and has been mainly based on the result of PD-L1 immunohistochemical staining. In the NCCN guideline, the FDA-approved 22C3 IHC assay for PD-L1 utilizes a cutoff of 50% tumor proportion score (TPS) for first-line therapy and 1% TPS for second-line therapy with pembrolizumab in NSCLC [1].

Recently, other biomarkers that aid prediction of response to immune checkpoint inhibitors such as tumor mutation burden (TMB) and microsatellite instability (MSI) have been emerging and PD-L1 expression alone is known to be an imperfect predictive biomarker. Despite the imperfect predictive value of PD-L1 expression alone, PD-L1 expression is the most widely used predictive marker in clinical practice to select patients most likely to respond the immunotherapy. Therefore, the identification and characterization of factors to predict patients with high expression of PD-L1 may still be significant. Various studies have reported the association between PD-L1 expression and driver genetic status in NSCLC, but the results have been conflicting and inconclusive [4,5,6,7,8,9,10,11,12,13,14]. A recent meta-analysis identified a high level of heterogeneity was observed between PD-L1 expression and driver genetic alterations [15]. Furthermore, immunotherapy is recommended as negative or unknown test results for *EGFR* mutations and *ALK* rearrangements in the NCCN guideline [1]. Therefore, there are limited and controversial data on PD-L1 expression in *EGFR*-mutated lung adenocarcinoma.

Therefore, we analyzed the association between PD-L1 expression and clinicopathological factors including driver genetic alterations to determine its predictive value. We also focused on PD-L1 expression in patients with *EGFR* mutated lung adenocarcinomas. We performed this analysis using a clinically validated PD-L1 IHC assay and clinically relevant cutoff values for PD-L1 positivity in a large single-institution study of surgically resected lung cancers with a high prevalence of *EGFR* mutation.

## 2. Results

### 2.1. Clinicopathological Characteristics of 1000 Surgically Resected Lung Cancers

The median age of the 1000 patients analyzed at diagnosis was 64 years (range, 24–89). Of all patients, 519 (51.9%) patients were <65 years old and 481 (48.1%) patients were ≥65 years old. Of the 1000 patients, 574 (57.4%) were male and 426 (42.6%) were female. Four-hundred-and-fifty-four patients (45.4%) had never smoked and 546 patients (54.6%) had a history of smoking. The pathological stage was I in 634 patients (63.4%), II in 176 patients (17.6%), III in 144 patients (14.4%), and IV in 30 patients (3%). The major histological types were adenocarcinomas (*n* = 773, 77.3%), followed by squamous cell carcinomas (*n* = 188, 18.8%), and the remaining types were in the minority. Of these 1000 patients, 65 (6.5%) received neoadjuvant chemoradiotherapy, 328 (32.8%) received adjuvant chemotherapy/radiation therapy, 31 (3.1%) patients were treated with EGFR tyrosine kinase inhibitors, and 2 (0.2%) patients were treated with an ALK inhibitor.

### 2.2. PD-L1 Expression in Lung Cancer

PD-L1 positivity is defined as proportion scores of 1% or higher and classified at 50% or higher on the basis of the clinical trial assay that may accurately predict the clinical response of patients with NSCLC treated with pembrolizumab. A total of 433 patients (43.3%) had PD-L1–positive lung cancer, including 200 (20.0%) with TPS of ≥50% and 233 (23.3%) with TPS of 1–49%. Of 785 adenocarcinomas, 290 (36.9%) were PD-L1 positive, including 177 (22.5%) with TPS of 1–49% and 113 (14.4%) with TPS of ≥50%. At the 1% and 50% cutoff value, PD-L1 expression rate was 36.9% and 14.4%, respectively. Of 188 squamous cell carcinomas, 136 (72.3%) were PD-L1 positive, including 50 (26.6%) with TPS of 1–49% and 86 (45.7%) with TPS of ≥50%. At 1% and 50% cut-off value, PD-L1 expression rate was 72.3% and 45.7%, respectively. Of 21 large cell neuroendocrine carcinomas, 6 (28.6%) were PD-L1 positive, including 5 (23.8%) with TPS of 1–49% and 1 (4.8%) with TPS of ≥50%. Two small cell carcinomas were all PD-L1 negative and only one case (25%) of four carcinoid tumors was PD-L1 positive.

### 2.3. Association between PD-L1 Expression Status and Clinicopathological Features in Lung Adenocarcinomas and Squamous Cell Carcinomas

In adenocarcinoma, PD-L1 expression was significantly associated with male gender, higher rate of smoking history, higher T, N stage, and AJCC stage at the 1% and 50% cutoff value for PD-L1 positivity (Table 1 and Table A1). PD-L1 expression at the 1% and 50% cutoff value showed higher expression in poorly differentiated histological variants, such as solid (1% cutoff value: 75.6%, 50% cutoff value: 51.1%), pleomorphic (87.5%, 87.5%), and cribriform predominant variant (52.4%, 23.8%) than any other predominant variants. We also analyzed whether there was a difference in the expression of PD-L1 even if the presence of a solid component was not predominant. At the 1% cutoff value for PD-L1 positivity, 62.9% (56/89) of solid component cases showed PD-L1 expression, and 27% (24/89) of a solid component cases showed PD-L1 expression at the 50% cutoff value. PD-L1 expression in the micropapillary predominant variant (53.1%) was significantly different in at the 1% cutoff value for PD-L1 positivity, but not different at the 50% cutoff value of PD-L1 positivity. PD-L1 expression rate was significantly low in well differentiated variants such as lepidic (14.9%, 0%), acinar predominant variants (29.6%, 8.7%), and invasive mucinous carcinoma (17.5%, 0%) at the 1% and 50% cutoff value for PD-L1 positivity. PD-L1 expression was not significantly associated with papillary predominant variant.

In squamous cell carcinoma, although poorly differentiated tumors showed higher PD-L1 expression, PD-L1 expression was not significantly associated with age, gender, smoking history, T, N stage, and AJCC stage at the 1% and 50% cutoff value for PD-L1 positivity (Table 2 and Table A2).

### 2.4. Association between PD-L1 Expression and Driver Genetic Status in Lung Adenocarcinoma

Of 785 adenocarcinomas, all the patients had their *EGFR* and *ALK* status identified. More than half of the patients (*n* = 424, 54.0%) had *EGFR* mutation. Among cases with *EGFR* mutation, 202 (47.6%) patients had L858R mutation in exon 21, 182 (42.9%) patients had exon 19 deletion, 17 (4.0%) patients had exon 20 insertion, and 23 (5.4%) patients had other less common mutations including L861Q, S768I, S719S, and S719A. Of 785 patients, 27 patients (3.4%) were *ALK* positive.

At the 1% cutoff value of PD-L1 positivity, PD-L1 expression was significantly higher in patients with wild-type EGFR compared to patients with EGFR mutants (46.0% vs. 29.2%, p < 0.001) (Table 3 and Table A3). Although there was no statistically significant difference, patients with exon19 deletions (34.6%) and exon20 insertion (35.3%) had higher PD-L1 expression than the patients with other EGFR mutations including such as the L858R mutation in exon21, and other mutations. At the 50% cutoff value of PD-L1 positivity, PD-L1 expression was also significantly higher in patients with wild-type EGFR than that in patients with EGFR mutant (21.9% vs. 8.0%, p < 0.001). Similar to the results for 1% cutoff value, patients with exon19 deletions (11.5%) and exon20 insertion (11.8%) had higher PD-L1 expression than the patients with other EGFR mutations. It is noteworthy that no statistically significant difference was observed. At the 1% cutoff value of PD-L1 positivity, ALK-positive patients had a higher PD-L1 expression than that in ALK-negative patients (81.5% vs. 35.4%, p < 0.001). Similar to the results for 1% cutoff value, PD-L1 expression at the 50% cutoff value was also significantly higher in ALK-positive patients than ALK-negative (48.1% vs. 13.2%, p < 0.001). Figure 1 shows representative cases of positive PD-L1 expression in ALK-positive lung adenocarcinoma.

### 2.5. Association between PD-L1 Expression Status and Clinicopathological Features in 424 EGFR-Mutated Lung Adenocarcinomas

We also analyzed the association between PD-L1 expression and clinicopathological factors in 424 *EGFR* mutated lung adenocarcinomas. Similar to the results found in the 785 lung adenocarcinomas, current or ex-smokers, higher T, N stage, AJCC stage, and poorly differentiated histological variants exhibited significantly higher PD-L1 expression at the 1% and 50% cut-off value for PD-L1 positive (Table 4 and Table A4).

### 2.6. Multivariate Logistic Regression Analysis of Clinicopathological Features for PD-L1 Expression in Lung Adenocarcinoma

In univariable logistic regression analysis to evaluate the association with PD-L1 positivity, we identified that males with higher rate of smoking history, higher T and N stage, histological solid pattern, *EGFR* wild type, and *ALK* positive were associated with PD-L1 expression with a 1% and 50% cutoff value (Table 5 and Table 6). In a multivariable logistic regression analysis to assess the independent association variables with PD-L1 positivity, higher rate of smoking history, higher N stage, histologic solid predominant pattern, *EGFR* wild type, and *ALK* positive remained significant predictors of PD-L1 expression with a 1% cutoff value (Figure 2a). At the 50% cutoff value, higher T and N stage, histological solid pattern, *EGFR* wild type, and *ALK* positive remained significant predictors of PD-L1 expression (Figure 2b).

## 3. Discussion

In this study, we identified PD-L1 expression with 1000 patient samples including 785 adenocarcinomas, 188 squamous cell carcinomas, 21 large cell neuroendocrine carcinomas, 4 carcinoid tumor, and 2 small cell carcinomas using the 22C3 PD-L1 assay. As expected, PD-L1 expression rate differed according to histological subtypes in this large cohort study. PD-L1 expression was significantly higher in squamous cell carcinoma than in adenocarcinoma, which was consistent with the previous studies [8,16,17]. At 1% cutoff value, PD-L1 expression rate was 36.9% and 72.3% in 785 lung adenocarcinomas and 188 squamous cell carcinomas, respectively. At 50% cutoff value, the PD-L1 expression rate was 14.4% and 45.7% in 785 adenocarcinomas and 188 squamous cell carcinomas. Although higher grade tumors were associated with higher PD-L1 expression, we did not find an association between PD-L1 expression and most of the clinicopathological factors in squamous cell carcinoma. This finding is similar to those from previous studies, wherein higher PD-L1 expression rate correlated only with squamous cell carcinoma, but not with the disease stage [17,18,19].

In adenocarcinoma, meanwhile, PD-L1 expression at the 1% and 50% cutoff value showed higher expression in poorly differentiated histologic variant, such as solid, pleomorphic, and cribriform predominant variants than any other predominant variants. We also identified that the expression of PD-L1 was significantly high even if a solid component was not predominantly present. Dong et al. [20] recently reported that either the solid predominant or component group showed significantly higher PD-L1 expression compared with the non-solid group. In their study, solid predominant types also had a high proportion of dual positive PD-L1 and tumor infiltrating lymphocytes (TIL), increased TMB, and higher frequency of GC > TA transversion [20]. These findings support the identification of solid variant adenocarcinoma with enhanced immunogenicity and association with good response to PD-1/PD-L1 inhibitors. Mucinous cribriform pattern was a previously reported histologic feature as a fusion gene-associated feature, such as *ALK*, *ROS1*, and *RET* rearranged lung cancer [21,22,23]. Since the cribriform pattern is closely associated with *ALK*-positive lung cancer, it was difficult to accurately determine which factors, predominantly contributed to the statistical significances.

In the NCCN guideline, immunotherapy is recommended as negative or unknown test results for *EGFR* mutations and *ALK* rearrangements [1]. Therefore, there are limited and controversial data on PD-L1 expression in *EGFR* mutated lung adenocarcinoma [24]. In the present study, the association between PD-L1 expression and genetic alterations was evaluated. At the 1% and 50% cutoff value of PD-L1 positivity, PD-L1 expression was significantly higher in patients with wild-type *EGFR* than that in patients with the *EGFR* mutant, which is consistent with several studies [8,25]. However, there have been conflicting results as to whether PD-L1 expression is associated with *EGFR* mutation. Several studies have identified that PD-L1 expression is higher in tumors with *EGFR* mutants [5,12], and other studies have reported that PD-L1 expression has no association with *EGFR* mutation [9,13]. In a meta-analysis study, a statistically significant negative correlation between PD-L1 expression and *EGFR* mutation in NSCLCs was recently identified, although a high level of heterogeneity was also observed [15].

The prevalence of *EGFR* mutations in NSCLC is much higher in East Asian countries (26–48%) compared with Western countries (10–20%) [26,27]. The incidence of *EGFR* mutations in patients with NSCLC was 54.0% (424 of 785) in our study. This frequency in Korea seemed to be particularly higher than in other East Asian countries, which is consistent with another Korean study [28]. The high incidence of *EGFR* mutation in our study might be due to inclusion of the enriched *EGFR* mutation cohort, such as Asian female, never-smoker patients. Therefore, the different frequency of *EGFR* mutation may be one of the reasons for the difference in PD-L1 expression according to their *EGFR* mutation status. Previous studies had analyzed a relatively smaller number of patients with *EGFR* mutations compared to our study and may support the conclusions that PD-L1 expression was significantly higher in patients with wild-type *EGFR* and with recognizing distinctive clinicopathologic features, especially solid histologic pattern and higher stage in *EGFR* mutant group.

Of note, at the 1% and 50% cutoff value of PD-L1 positivity, *ALK*-positive patients had higher PD-L1 expression than *ALK*-negative patients. Up-regulation of PD-L1 expression as a result of constitutive oncogenic signaling had also been reported in NSCLC harboring *EML4*–*ALK* rearrangements [10]. However, it has been reported that PD-L1 expression alone in *ALK*-positive patients does not predict response to the immunotherapy [29]. Despite PD-L1 expression, these patients showed relatively low response rate to PD-1/PD-L1 inhibitors [29]. This could be explained by the low rates of PD-L1 expression and high levels of CD8+ TILs, because of a lack of effector cells to function an antitumor immune response [28]. We unfortunately restricted our analysis to PD-L1 expression, and not CD8- + TILs. A large cohort study, therefore, needs to be performed to validate these findings.

Meanwhile, the association between PD-L1 expression and *ALK* rearrangement was also conflicting in previous studies. Most of studies have reported that PD-L1 expression has no association with *ALK* status [4,6,7,8,11,12,14,30]. In a meta-analysis study, no statistically significant difference in PD-L1 expression was found among NSCLCs with different *ALK* status [15]. Table 7 presents a summary of cases showing the association between PD-L1 expression and *ALK* positive in published literatures. As shown in Table 7, most of studies except one by Koh et al. [9]. identified no significant association with between *ALK* status and PD-L1 expression. Incecco et al. [5]. reported that PD-L1 expression levels were high in patients with *ALK* translocations, although the association was not statistically significant.

These discrepancies might be attributed to different ethnic characteristics, the use of various antibody clones, different IHC protocols and platform, different cutoff values for PD-L1 positivity, and different types of specimens (tissue microarray (TMA) vs. whole tumor section). Currently, multiple PD-L1 IHC assays are used to determine the expression of PD-L1 in lung cancer. Each PD-L1 IHC assay is linked to a specific therapeutic agent. Different PD-L1 IHC assays including different antibody clones, and different platforms have been developed and approved in parallel with different therapeutic agents, with different cutoffs determined by clinical response. In previously published literature, the PD-L1 antibodies used had not been thoroughly validated, leading to conflicting results regarding PD-L1 expression. We performed IHC for the PD-L1 expression using a 22C3 assay as the companion diagnosis assay at the 1% and 50% cutoff value, used in pembrolizumab trials [2]. The potential difference in results between the whole tumor section and TMA may be one of the reasons of discrepancies [31]. PD-L1 expression in NSCLC showed a high discordance rate between TMA samples and whole tumor section [32,33]. TMA may not be representative of whole tumor tissue due to the heterogeneity of PD-L1 expression. Especially, it makes difficult to assess the PD-L1 positivity in adenocarcinoma showing *ALK*-positive-associated histologic features such as mucinous cribriform pattern. PD-L1 staining was more heterogenous and weak due to abundant mucinous components. Although intratumoral heterogeneity of PD-L1 expression and interassay variation may contribute to the conflicting results in terms of association with genomic status, our results are adequately reliable for the reasons mentioned above.

Therefore, our results are useful to guide the selection of lung cancer patients with PD-L1 expression for further immunotherapy. Patients with wild type of *EGFR* and *ALK*-positive lung adenocarcinoma may represent a potential selective group, which has a better chance of good response to immunotherapy.

One major limitation of this study is that it only includes recent retrospectively collected cases and survival information is therefore not yet available. A second limitation is the lack of data available for response to PD-1/PD-L1 inhibitors in this patient population. Our results should be further validated by a study of the clinical trial group treated with immunotherapy. 

To the best of our knowledge, this is the largest study to evaluate the association between PD-L1 expression and clinicopathological and molecular features in lung cancer. In adenocarcinoma, PD-L1 expression using 22C3 assay at clinically relevant cutoff could be predicted in higher N stage, histologically solid pattern, wild-type *EGFR*, and *ALK* positive in lung adenocarcinoma.

## 4. Materials and Methods

### 4.1. Patients

This study was approved by the Institutional Review Board (IRB) of Samsung Medical Center, Seoul, Korea (IRB File No. 2019-07-047-003, 22 July 2019). This study included 1000 patient samples including 785 adenocarcinomas, 188 squamous cell carcinomas, 21 large cell neuroendocrine carcinomas, 4 carcinoid tumor, and 2 small cell carcinomas who underwent lung resection for lung mass between July 2017 and March 2019 at Samsung Medical Center. Patients who were histologically confirmed to have lung cancer and had sufficient tissue for PD-L1 immunohistochemistry (IHC) staining were considered eligible for the study.

Clinicopathological features, including sex, age, smoking history, histological subtypes, differentiation, pathologic stage, and molecular genotype were retrospectively obtained from medical records. Pathologic stage was defined using the American Joint Committee on Cancer Staging Manual (AJCC), eighth edition. Histological subtypes of lung adenocarcinoma were classified according to the new International Association for the Study of Lung Cancer/American Thoracic Society/European Respiratory Society (IASLC/ATS/ ERS) multidisciplinary classification of lung adenocarcinoma. We recorded the predominant histological pattern (lepidic, acinar, papillary, micropapillary, and solid), which can be associated with prognosis.

### 4.2. PD-L1 Immunohistochemistry

PD-L1 expression was assessed in formalin-fixed paraffin embedded tumor samples acquired by surgical resection sample from each patients, using the PD-L1 IHC 22C3 pharmDx assay (Dako North America, Carpinteria, CA, USA) according to manufacturer’s instructions [36]. For the 22C3 pharmDx assay, sections were stained with anti-PD-L1 22C3 mouse monoclonal primary antibody using the EnVision FLEX visualization system on a Dako Autostainer Link 48 system with negative reagent controls and cell line run controls, as described in the PD-L1 IHC 22C3 pharmDx package insert [37]. Deparaffinization, rehydration, and target retrieval was performed with a 3-in-1 procedure using PT Link. Following peroxidase blocking, specimens were incubated with monoclonal mouse primary antibody to PD-L1 or the negative control reagent. Specimens were then incubated with Mouse Linker, followed by incubation with a ready-to-use visualization reagent consisting of secondary antibody molecules and horseradish peroxidase molecules coupled to a dextran polymer backbone. The enzymatic conversion of the subsequently added chromogen results in the precipitation of a visible reaction product at the site of the antigen. The color of the chromogenic reaction is modified using a chromogen enhancement reagent; the specimen may then be counterstained and cover slipped. Results were interpreted using a light microscope [38]. All stained slides were evaluated by a board-certified pathologist for PD-L1 membrane staining. The percentage of membranous stained tumor cells in the overall area of the tumor (tumor proportion score) was scored regardless of intensity [2]. Positivity was evaluated by two different cut-off values, 1% and 50%, based on cutoffs used in pembrolizumab clinical trials [2,36].

### 4.3. Analysis of EGFR Mutation

*EGFR* gene alteration was detected by either real-time PCR with PNA-clamping methods, direct sequencing, or both methods. The PNA-ClampTM *EGFR* mutation detection kit (PANAGENE, Inc., Daejeon, Korea) was used for real-time PCR, performed as described previously [39]. When detection was performed only with direct sequencing, exon 18, 19, 20, and 21 were sequenced as previously described [40]. When both methods were used, exons containing mutations detected by real-time PCR were sequenced, and exon 19 was sequenced if no mutation was detected by real-time PCR.

### 4.4. Analysis of ALK Fusion

For the *ALK* fusion, ALK immunohistochemistry was performed using an anti-ALK mouse monoclonal antibody (clone: 5A4, Leica Biosystems Newcastle Ltd., UK; diluted 1:50) and the Leica Bond III automated system (Leica Biosystems Melbourne Pty Ltd.) in 785 lung adenocarcinomas. The sections were incubated at pH 9 for 30 min at 100 °C. The *ALK* fusions in the 33 ALK-immunohistochemically positive cases were confirmed by fluorescence in situ hybridization (FISH). *ALK* FISH testing was performed using the Vysis *ALK* BreakApart probe kit (Abbott Molecular, Des Plaines, IL, USA) and a positive FISH result for *ALK* rearrangement was defined as >15% of tumor cells with a split signal.

### 4.5. Statistical Analysis

The association between PD-L1 expression and clinicopathological features was analyzed with the χ2 test or Fisher’s exact test. The association between clinicopathological features or genetic alterations with PD-L1 positivity was also assessed by univariable and multivariable logistic regression models. A *p*-value of less than 0.05 was considered to indicate a statistically significant difference. All analyses were carried out using SPSS version 22 (SPSS Inc, Chicago, IL, USA).

## 5. Conclusions

In conclusion, PD-L1 expression using a 22C3 assay at clinically relevant cutoff could be predicted in higher N stage, histological solid pattern, wild-type *EGFR*, and *ALK* positive in lung adenocarcinoma. These results may be beneficial for selecting of high-risk patients for good response to the immunotherapy.

## Figures and Tables

**Figure 1 ijms-20-04794-f001:**
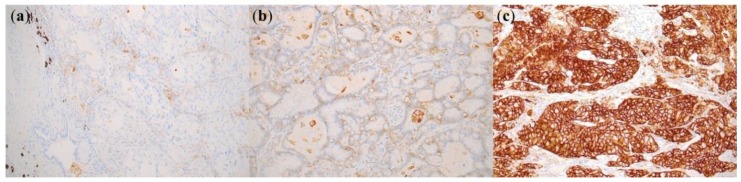
Representative cases of PD-L1 staining (22C3) observed in *ALK*-positive lung adenocarcinomas. (**a**) PD-L1 negative (200×); (**b**) PD-L1 weak positive (>50%) (200×); (**c**) PD-L1 strong positive (>50%) (200×).

**Figure 2 ijms-20-04794-f002:**
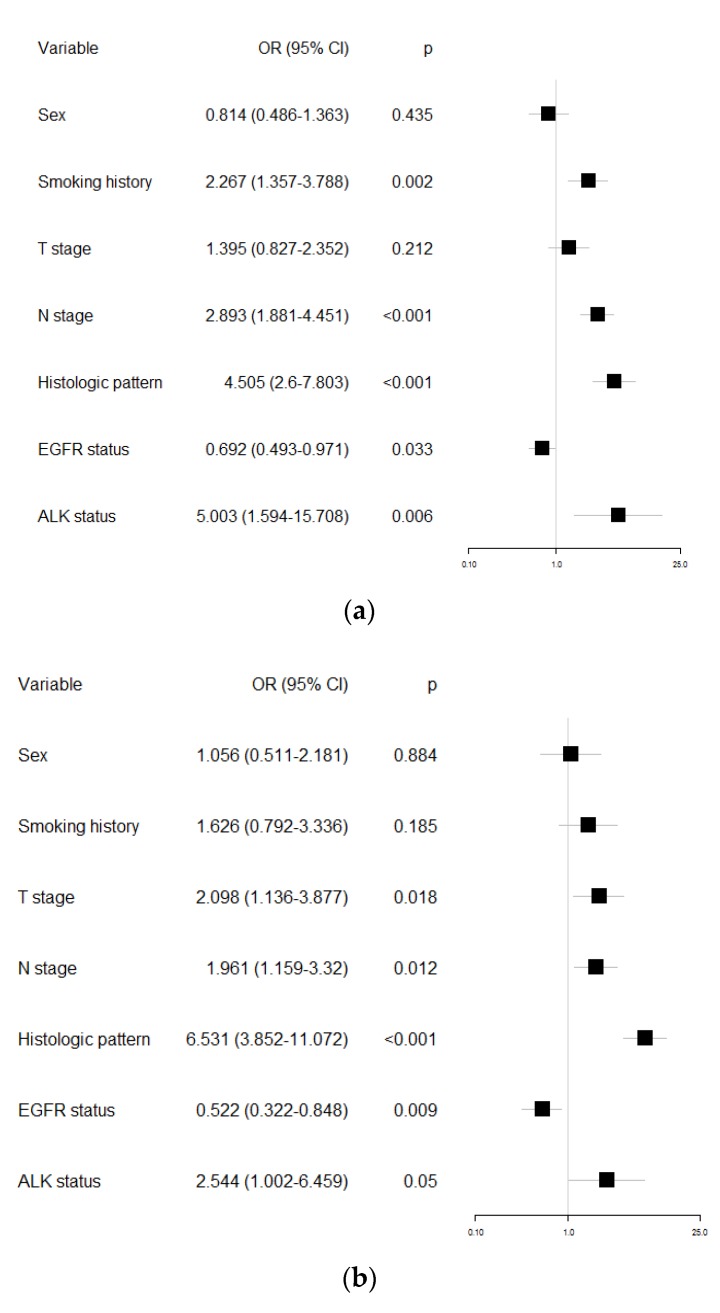
Forest plots reflecting the results of multivariate logistic regression analysis of clinicopathological features for PD-L1 expression in lung adenocarcinoma: (**a**) 1% cutoff value for PD-L1 positivity; (**b**) 50% cutoff value for PD-L1 positivity.

**Table 1 ijms-20-04794-t001:** Association between programmed cell death ligand 1 (PD-L1) expression status and clinicopathological features in 785 lung adenocarcinomas.

			Adenocarcinoma (*n* = 785)			
			PD-L1 Expression			
Variable		Total Patients No.	<1% (*n* = 495, 63.1%)	1–49% (*n* = 177, 22.5%)	≥50% (*n* = 113, 14.4%)	*p*-Value
Age						0.173
	<65 year	440 (56.1%)	290 (65.9%)	91 (20.7%)	59 (13.4%)	
	≥65 year	345 (43.9%)	205 (59.4%)	86 (24.9%)	54 (15.7%)	
Sex						<0.001
	Male	377 (48%)	213 (56.5%)	92 (24.4%)	72 (19.1%)	
	Female	408 (52%)	282 (69.1%)	85 (20.8%)	41 (10.0%)	
Smoking status						<0.001
	Never smoker	438 (55.8%)	308 (70.3%)	87 (19.9%)	43 (9.8%)	
	Current or Ex smoker	347 (44.2%)	187 (53.9%)	90 (25.9%)	70 (20.2%)	
T_stage						0.001
	1	519 (66.3%)	354 (68.2%)	109 (21.0%)	56 (10.8%)	
	2	178 (22.7%)	96 (53.9%)	48 (27.0%)	34 (19.1%)	
	3	68 (8.7%)	36 (52.9%)	15 (22.1%)	17 (25.0%)	
	4	18 (2.3%)	8 (44.4%)	4 (22.2%)	6 (33.3%)	
N_stage						<0.001
	0	629 (82.1%)	427 (67.9%)	125 (19.9%)	77 (12.2%)	
	1	37 (4.8%)	15 (40.5%)	15 (40.5%)	7 (18.9%)	
	2	100 (13.1%)	41 (41.0%)	31 (31.0%)	28 (28.0%)	
M_stage						0.049
	0	752 (96.2%)	477 (63.4%)	164 (21.8%)	111 (14.8%)	
	1	30 (3.8%)	16 (53.3%)	12 (40.0%)	2 (6.7%)	
AJCC stage						<0.001
	1	544 (70.3%)	377 (69.3%)	108 (19.9%)	59 (10.8%)	
	2	99 (12.8%)	56 (56.6%)	23 (23.2%)	20 (20.2%)	
	3	103 (13.3%)	41 (39.8%)	31 (30.1%)	31 (30.1%)	
	4	28 (3.6%)	14 (50.0%)	12 (42.9%)	2 (7.1%)	
Preop Tx.						0.011
	No	739 (94.1%)	471 (63.7%)	168 (22.7%)	100 (13.5%)	
	PreopCCRT	39 (5.0%)	19 (48.7%)	7 (17.9%)	13 (33.3%)	
	PreopChemo	7 (0.9%)	5 (71.4%)	2 (28.6%)	0	
Vascular invasion						0.002
	No	738 (95.3%)	475 (64.4%)	161 (21.8%)	102 (13.8%)	
	YES	36 (4.7%)	13 (36.1%)	12 (33.3%)	11 (30.6%)	
Lymphatic invasion						<0.001
	No	608 (78.6%)	413 (67.9%)	120 (19.7%)	75 (12.3%)	
	YES	166 (21.4%)	75 (45.2%)	53 (31.9%)	38 (22.9%)	
Differentiation						<0.001
	W/D	41 (5.2%)	37 (90.2%)	4 (9.8%)	0	
	M/D	561 (71.5%)	385 (68.6%)	126 (22.5%)	50 (8.9%)	
	P/D	147 (18.7%)	47 (32.0%)	40 (27.2%)	60 (40.8%)	
Predominant pattern						<0.001
	Lepidic	47 (6.1%)	40 (85.1%)	7 (14.9%)	0	
	Acinar	402 (52.5%)	283 (70.4%)	84 (20.9%)	35 (8.7%)	
	Papillary	108 (14.1%)	69 (63.9%)	29 (26.9%)	10 (9.3%)	
	Micropapillary	49 (6.4%)	23 (46.9%)	17 (34.7%)	9 (18.4%)	
	Solid	90 (11.8%)	22 (24.4%)	22 (24.4%)	46 (51.1%)	
	Cribriform	21 (2.7%)	10 (47.6%)	6 (28.6%)	5 (23.8%)	
	Mucinous	40 (5.2%)	33 (82.5%)	7 (17.5%)	0	
	Pleomorphic	8 (1.0%)	1 (12.5%)	0	7 (87.5%)	

Abbreviations: CCRT, concurrent chemoradiotherapy; W/D, well differentiated; M/D, moderately differentiated; P/D, poorly differentiated.

**Table 2 ijms-20-04794-t002:** Association between PD-L1 expression status and clinicopathological features in 188 lung squamous cell carcinoma.

			Squamous Cell Carcinoma (*n* = 188)			
			PD-L1 Expression			
Variable		Total Patients No.	<1% (*n* = 52, 27.7%)	1–49% (*n* = 50, 26.6%)	≥50% (*n* = 86, 45.7%)	*p*-Value
Age						0.927
	<65	71 (37.8%)	19 (26.8%)	20 (28.2%)	32 (45.1%)	
	≥65	117 (62.2%)	33 (28.2%)	30 (25.6%)	54 (46.2%)	
Sex						
	Male	174 (92.6%)	50 (28.7%)	47 (27.0%)	77 (44.3%)	
	Female	14 (7.4%)	2 (14.3%)	3 (21.4%)	9 (64.3%)	
Smoking status						0.944
	Never smoker	13 (6.9%)	4 (30.8%)	3 (23.1%)	6 (46.2%)	
	Current or Ex smoker	175 (93.1%)	48 (27.4%)	47 (26.9%)	80 (45.7%)	
T_stage						0.698
	1	73 (38.9%)	20 (27.4%)	20 (27.4%)	33 (45.2%)	
	2	54 (28.7%)	17 (31.5%)	14 (25.9%)	23 (42.6%)	
	3	45 (23.9%)	11 (24.4%)	10 (22.2%)	24 (53.3%)	
	4	15 (8.0%)	3 (20.0%)	6 (40.0%)	6 (40.0%)	
N_stage						0.449
	0	125 (68.3%)	32 (25.6%)	38 (30.4%)	55 (44.0%)	
	1	33 (18%)	7 (21.2%)	7 (21.2%)	19 (57.6%)	
	2	25 (13.7%)	9 (36.0%)	5 (20.0%)	11 (44.0%)	
M_stage						0.086
	0	183 (97.9%)	48 (26.2%)	50 (27.3%)	85 (46.4%)	
	1	4 (2.1%)	3 (75.0%)	0	1 (25.0%)	
AJCC stage						0.063
	1	77 (41.8%)	23 (29.9%)	26 (33.8%)	28 (36.4%)	
	2	68 (37%)	14 (20.6%)	16 (23.5%)	38 (55.9%)	
	3	37 (20.1%)	10 (27.0%)	8 (21.6%)	19 (51.4%)	
	4	2 (1.1%)	2 (100%)	0	0	
Preop Tx.						0.343
	No	169 (89.9%)	45 (26.6%)	48 (28.4%)	76 (45.0%)	
	PreopCCRT	16 (8.5%)	5 (31.3%)	2 (12.5%)	9 (56.3%)	
	PreopChemo	3 (1.6%)	2 (66.7%)	0	1 (33.3%)	
Vascular invasion						0.247
	No	172 (93%)	45 (26.2%)	49 (28.5%)	78 (45.3%)	
	YES	13 (7%)	5 (38.5%)	1 (7.7%)	7 (53.8%)	
Lymphatic invasion						0.139
	No	144 (77.8%)	34 (23.6%)	40 (27.8%)	70 (48.6%)	
	YES	41 (22.2%)	16 (39.0%)	10 (24.4%)	15 (36.6%)	
Differentiation						0.044
	W/D	9 (4.8%)	2 (22.2%)	4 (44.4%)	3 (33.3%)	
	M/D	129 (68.6%)	29 (22.5%)	35 (27.1%)	65 (50.4%)	
	P/D	41 (21.8%)	15 (36.6%)	11 (26.8%)	15 (36.6%)	

Abbreviations: CCRT, concurrent chemoradiotherapy; W/D, well differentiated; M/D, moderately differentiated; P/D, poorly differentiated.

**Table 3 ijms-20-04794-t003:** Association between PD-L1 expression and driver genetic status in lung adenocarcinoma.

				Adenocarcinoma (*n* = 785)			
				PD-L1			
Driver Gene			Total Patients No.	<1% (*n* = 495)	1–49% (*n* = 177)	≥50% (*n* = 113)	*p*-Value
*EGFR*							<0.001
	Wild		361 (46.0%)	195 (54.0%)	87 (24.1%)	79 (21.9%)	
	Mutant		424 (54.0%)	300 (70.8%)	90 (21.2%)	34 (8.0%)	
		Exon19del	182 (42.9%)	119 (65.4%)	42 (23.1%)	21 (11.5%)	
		L858R	202 (47.6%)	153 (75.7%)	39 (19.3%)	10 (5.0%)	
		Exon20ins	17 (4.0%)	11 (64.7%)	4 (23.5%)	2 (11.8%)	
		Others	23 (5.4%)	17 (73.9%)	5 (21.7%)	1 (4.3%)	
*ALK*							<0.001
	Wild		758 (96.6%)	490 (64.6%)	168 (22.2%)	100 (13.2%)	
	Mutant		27 (3.4%)	5 (18.5%)	9 (33.3%)	13 (48.1%)	

**Table 4 ijms-20-04794-t004:** Association between PD-L1 expression status and clinicopathological features in 424 *EGFR* mutated lung adenocarcinoma.

				*EGFR*-Mutated Adenocarcinoma (*n* = 424)		
				PD-L1 Expression		
Variable		Total Patients No.	<1% (*n* = 300, 70.8%)	1–49% (*n* = 90, 21.2%)	≥50% (*n* = 34, 8.0%)	*p*-Value
Age						0.159
	<65	255 (60.1%)	188 (73.7%)	51 (20.0%)	16 (6.3%)	
	≥65	169 (39.9%)	112 (66.3.%)	39 (23.1%)	18 (10.7%)	
Sex						0.202
	Male	157 (37.0%)	103 (65.6%)	39 (24.8%)	15 (9.6%)	
	Female	267 (63.0%)	197 (73.8%)	51 (19.1%)	19 (7.1%)	
Smoking status						0.025
	Never smoker	279 (65.8%)	209 (74.9%)	49 (17.6%)	21 (7.5%)	
	Current or Ex smoker	145 (34.2%)	91 (62.8%)	41 (28.3%)	13 (9.0%)	
T_stage						0.003
	1	292 (69.2%)	223 (76.4%)	53 (18.2%)	16 (5.5%)	
	2	96 (22.7%)	53 (55.2%)	30 (31.3%)	13 (13.5%)	
	3	30 (7.1%)	20 (66.7%)	5 (16.7%)	5 (16.7%)	
	4	4 (0.9%)	3 (75.0%)	1 (25.0%)	0.00%	
N_stage						0.001
	0	341 (83.2%)	256 (75.1%)	64 (18.8%)	21 (6.2%)	
	1	18 (4.4%)	11 (61.1%)	5 (27.8%)	2 (11.1%)	
	2	51 (12.4%)	25 (49.0%)	16 (31.4%)	10 (19.6%)	
M_stage						0.011
	0	406 (96.2%)	291 (71.7%)	81 (20.0%)	34 (8.4%)	
	1	16 (3.8%)	8 (50.0%)	8 (50.0%)	0	
AJCC stage						<0.001
	1	315 (75.7%)	238 (75.6%)	58 (18.4%)	19 (6.0%)	
	2	37 (8.9%)	26 (70.3%)	8 (21.6%)	3 (8.1%)	
	3	50 (12.0%)	25 (50.0%)	14 (28.0%)	11 (22.0%)	
	4	14 (3.4%)	6 (42.9%)	8 (57.1%)	0	
Preop Tx.						0.298
	No	407 (96.0%)	284 (69.8%)	89 (21.9%)	34 (8.4%)	
	PreopCCRT	12 (2.8%)	11 (91.7%)	1 (8.3%)	0	
	PreopChemo	5 (1.2%)	5 (100%)	0	0	
Vascular invasion						0.161
	No	398 (96.1%)	286 (71.9%)	80 (20.1%)	32 (8.0%)	
	YES	16 (3.9%)	8 (50.0%)	6 (37.5%)	2 (12.5%)	
Lymphatic invasion						<0.001
	No	323 (78.0%)	250 (77.4%)	57 (17.6%)	16 (5.0%)	
	YES	91 (22.0%)	44 (48.4%)	29 (31.9%)	18 (19.8%)	
Differentiation						<0.001
	W/D	20 (4.7%)	18 (90.0%)	2 (10.0%)	0	
	M/D	340 (80.2%)	251 (73.8%)	70 (20.6%)	19 (5.6%)	
	P/D	51 (12.0%)	20 (39.2%)	16 (31.4%)	15 (29.4%)	
Predominant pattern						<0.001
	Lepidic	26 (6.3%)	23 (88.5%)	3 (11.5%)	0	
	Acinar	273 (65.8%)	205 (75.1%)	54 (19.8%)	14 (5.1%)	
	Papillary	58 (14.0%)	38 (65.5%)	16 (27.6%)	4 (6.9%)	
	Micropapillary	27 (6.5%)	13 (48.1%)	9 (33.3%)	5 (18.5%)	
	Solid	21 (5.1%)	7 (33.3%)	5 (23.8%)	9 (42.9%)	
	Cribriform	6 (1.4%)	3 (50.0%)	1 (16.7%)	2 (33.3%)	
	Mucinous	4 (1.0%)	4 (100.0%)	0	0	
*EGFR* mutation genotyping						0.229
	Exon19del	182 (42.9%)	119 (65.4%)	42 (23.1%)	21 (11.5%)	
	L858R	202 (47.6%)	153 (75.7%)	39 (19.3%)	10 (5.0%)	
	Exon20insertion	17 (4.0%)	11 (64.7%)	4 (23.5%)	2 (11.8%)	
	Others	23 (5.4%)	17 (73.9%)	5 (21.7%)	1 (4.3%)	

Abbreviations: CCRT, concurrent chemoradiotherapy; W/D, well differentiated; M/D, moderately differentiated; P/D, poorly differentiated.

**Table 5 ijms-20-04794-t005:** Multivariate logistic regression analysis of clinicopathological features for PD-L1 expression in lung adenocarcinoma (1% cutoff value for PD-L1 positivity).

Factors to Predict PD-L1 Positivity in Lung Adenocarcinoma		Univariable Analysis		Multivariable Analysis	
(1% Cutoff Value for PD-L1 Positivity)		OR (95% CI)	*p*-Value	OR (95% CI)	*p*-Value
Sex	Male (vs. Female)	1.723 (1.286–2.309)	<0.001	0.814 (0.486–1.363)	0.435
Age	≥65 (vs. < 65)	1.320 (0.986–1.767)	0.062		
Smoking history	Ever (vs. Never) smoker	2.027 (1.510–2.721)	<0.001	2.267 (1.357–3.788)	0.002
T stage	T stage 3/4 (vs. Stage 1/2)	1.739 (1.108–2.728)	0.016	1.395 (0.827–2.352)	0.212
N stage	N stage 1/2 (vs. N stage 0)	3.058 (2.092–14.69)	<0.001	2.893 (1.881–4.451)	<0.001
M stage	M1 stage (vs. M0 stage)	1.518 (0.730–3.157)	0.264		
Histologic pattern	Solid (vs. Non-solid) variant	6.568 (3.955–10.907)	<0.001	4.505 (2.6–7.803)	<0.001
*EGFR* status	Mutant (vs. Wild) type	0.521 (0.388–0.698)	<0.001	0.692 (0.493–0.971)	0.033
*ALK* status	Positive (vs. Negative)	8.05 (3.012–21.486)	<0.001	5.003 (1.594–15.708)	0.006

**Table 6 ijms-20-04794-t006:** Multivariate logistic regression analysis of clinicopathological features for PD-L1 expression in lung adenocarcinoma (50% cutoff value for PD-L1 positivity).

Factors to Predict PD-L1 Positivity in Lung Adenocarcinoma		Univariable Analysis		Multivariable Analysis	
(50% Cutoff Value for PD-L1 Positivity)		OR (95% CI)	*p*-Value	OR (95% CI)	*p*-Value
Sex	Male (vs. Female)	2.135 (1.413–3.226)	<0.001	1.056 (0.511–2.181)	0.884
Age	≥65 (vs. < 65)	1.204 (0.808–1.796)	0.362		
Smoking history	Ever (vs. Never) smoker	2.335 (1.550–3.518)	<0.001	1.626 (0.792–3.336)	0.185
T stage	T stage 3/4 (vs. Stage 1/2)	2.490 (1.469–4.218)	0.001	2.098 (1.136–3.877)	0.018
N stage	N stage 1/2 (vs. N stage 0)	2.500 (1.590–3.932)	<0.001	1.961 (1.159–3.320)	0.012
M stage	M1 stage (vs. M0 stage)	0.461 (0.108–1.972)	0.296		
Histologic pattern	Solid (vs. Non-solid) variant	9.839 (6.044–16.015)	<0.001	6.531 (3.852–11.072)	<0.001
*EGFR* status	Mutant (vs. Wild) type	0.346 (0.227–0.527)	<0.001	0.522 (0.322–0.848)	0.009
*ALK* status	Positive (vs. Negative)	6.064 (2.769–13.277)	<0.001	2.544 (1.002–6.459)	0.05

**Table 7 ijms-20-04794-t007:** Summary of cases showing the association between PD-L1 expression and *ALK* positive in published literatures.

No. of Study	Authors	Years	Ethnicity	Tissue Type	PD-L1 Antibody	Cutoff Value	*ALK* Positive Cases	PD-L1 Positive Cases	*p*-Value
1 [34]	Zhang	2014	East-Asian	Whole tissue section	SAB2900365	40%	9/143 (6.3%)	3/9 (33.3%)	0.494
2 [13]	Yang	2014	East-Asian	Whole tissue section	Proteintech Group Inc.	5%	3/163 (1.84%)	2/3 (66.7%)	0.564
3 [4]	Cooper	2015	Non-East-Asian	TMA	22C3	50%	3/678 (0.44%)	0/3 (0%)	1
4 [5]	Incecco	2015	Non-East-Asian	Whole tissue section	ab58810	5%	10/125 (8%)	3/10 (30%)	1
5 [9]	Koh	2015	East-Asian	TMA	E1L3N	10%	23/497 (4.63%)	18/23 (78.3%)	0.054
6 [10]	Ota	2015	East-Asian	Whole tissue section	Lifespan Biosciences	H-score	11/134 (8.21%)	N/A	<0.001
7 [12]	Song	2016	East-Asian	Whole tissue section	Proteintech Group Inc.	5%	18/385 (4.68%)	10/18 (55.6%)	0.53
8 [6]	Inamura	2016	East-Asian	TMA	E1L3N	5%	10/268 (3.73%)	1/10 (10%)	1
9 [7]	Inoue	2016	East-Asian	TMA	E1L3N	5%	10/654 (1.53%)	5/10 (50%)	0.169
10 [30]	Huynh	2016	Non-East-Asian	TMA	E1L3N	5%	4/261 (1.53%)	1/4 (25%)	N/A
11 [11]	Rangachari	2017	Non-East-Asian	Whole tissue section	22C3	50%	4/71 (5.63%)	1/3 (33.3%)	N/A
12 [35]	Jia	2018	East-Asian	Whole tissue section	E1L3N	10%	5/55 (9.09%)	1/5 (20%)	0.822
13 [8]	Kim	2018	East-Asian	TMA	22C3	1%, 50%	24/429 (5.59%)	3/24 (12.5%)	>0.05
14	This study	2019	East-Asian	Whole tissue section	22C3	1%, 50%	27/994 (2.72%)	22/27 (81.5%)	<0.001

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
