# Peer review of "Association with PD-L1 Expression and Clinicopathological Features in 1000 Lung Cancers: A Large Single-Institution Study of Surgically Resected Lung Cancers with a High Prevalence of *EGFR* Mutation"

_ijms, 2019, doi:10.3390/ijms20194794_

Round 1

Reviewer 1 Report

The authors of manuscript entitled “Association with PD-L1 expression and clinicopathological features in 1000 lung cancers: A large single-institution study of surgically resected lung cancers with a high prevalence of EGFR mutation” enrolled 1000 lung cancer patients (from July 2017 to March 2019) with their resected tumor samples and clinical information. Immunohistochemistry was performed to examine the expression levels of PD-L1 and the mutation status of EGFR and ALK fusion were also revealed. The authors claimed that PD-L1 was highly expression in squamous cell carcinoma than it in adenocarcinoma. In adenocarcinoma, PD-L1 expression was associated with lymph node metastasis and solid histological pattern. In addition, PD-L1 expression was associated with several clinical features in the EGFR-mutation patients. This reviewer appreciate the high patient number in this study. However, the data were poorly processed. The number within a table or between related tables are not matched. Accordingly, the results of statistical analysis is doubtful. In addition, there are too many tables presented in the manuscript and some tables were not well organized and described. Finally, the conclusion made by the authors is blurred. Other comments are listed as following.

Major comments:

In the Results section, line 82-83, there are 433 patients with PD-L1-positive lung cancer, including 233 with TPS >= 50% and 139 with TPS of 1%-49%. Do the patient number correct? Similar unmatched number also showed in Table 1. The sum of total number in almost all variables was not 785.

In the head of Table 2 and Table 3, for the classification of PD-L1 expression, is it >1% or <1%?

Higher magnification of the representative IHC images in Figure 1 should be provided. It is impossible to see the membrane staining pf PD-L1 in the representative images. Why did the authors show the IHC images of ALK-positive lung cancer, while only 27 patients carried ALK fusion? The representative images for PD-L1-negative (<1%) should also be shown.

Since the authors claimed that EGFR wild type may be a predictor for PD-L1 expression (Figure 2), why did the authors only perform statistical analysis for the patients with EGFR mutations (Table 4)?

Minor comments:

The image quality of Figure 2 is low. What is the P value for the multivariate logistic regression analysis for individual variables? The number of % for the PD-L1-positive neuroendocrine carcinoma in missing. line 88 A typo in line 93 “Supplementary Table 1”

Line 116-line 118 is redundant. The same sentences showed in page 3, line 109-line 111.

The titles of Tables 5, 6 and 7 are the same.

    6. The Discussion is good.

Author Response

Reviewer(s)' Comments to Author: 

Reviewer 1 
The authors of manuscript entitled “Association with PD-L1 expression and clinicopathological features in 1000 lung cancers: A large single-institution study of surgically resected lung cancers with a high prevalence of EGFR mutation” enrolled 1000 lung cancer patients (from July 2017 to March 2019) with their resected tumor samples and clinical information. Immunohistochemistry was performed to examine the expression levels of PD-L1 and the mutation status of EGFR and ALK fusion were also revealed. The authors claimed that PD-L1 was highly expression in squamous cell carcinoma than it in adenocarcinoma. In adenocarcinoma, PD-L1 expression was associated with lymph node metastasis and solid histological pattern. In addition, PD-L1 expression was associated with several clinical features in the EGFR-mutation patients. This reviewer appreciate the high patient number in this study. However, the data were poorly processed. The number within a table or between related tables are not matched. Accordingly, the results of statistical analysis is doubtful. In addition, there are too many tables presented in the manuscript and some tables were not well organized and described. Finally, the conclusion made by the authors is blurred. Other comments are listed as following.

Major comments:

In the Results section, line 82-83, there are 433 patients with PD-L1-positive lung cancer, including 233 with TPS >= 50% and 139 with TPS of 1%-49%. Do the patient number correct? Similar unmatched number also showed in Table 1. The sum of total number in almost all variables was not 785.

Response: As the reviewer rightly pointed out, we have changed the unmatched number in result section and Tables in revised manuscript. The sum of total number in several variables such as lymphatic and vascular invasion was not 785. It only included recent retrospectively collected cases. Therefore, these variables were not currently available due to the lack of data

“A total of 433 patients (43.3%) had PD-L1–positive lung cancer, including 200 (20.0%) with TPS of ≥ 50% and 233 (23.3%) with TPS of 1%-49%. Of 785 adenocarcinomas, 290 (36.9%) were PD-L1 positive, including 177 (22.5%) with TPS of 1-49% and 113 (14.4%) with TPS of ≥ 50%.”

In the head of Table 2 and Table 3, for the classification of PD-L1 expression, is it >1% or <1%?

Response: We have changed into <1%. Table 2 and 3 represent the association between PD-L1 expression and clinicopathological features according to 1% and 50% cutoff value.

Higher magnification of the representative IHC images in Figure 1 should be provided. It is impossible to see the membrane staining pf PD-L1 in the representative images. Why did the authors show the IHC images of ALK-positive lung cancer, while only 27 patients carried ALK fusion? The representative images for PD-L1-negative (<1%) should also be shown.

Response: In our study, we demonstrated that ALK-positive patients had higher PD-L1 expression than ALK-negative patients at the 1% and 50% cutoff value of PD-L1 positivity. However, the association between PD-L1 expression and ALK rearrangement was also conflicting in previous studies. Most of studies have reported that PD-L1 expression has no association with ALK status. We mentioned these discrepancies in discussion and added another reason in revised manuscript.

“It makes difficult to assess the PD-L1 positivity in adenocarcinoma showing ALK positive-associated histologic features such as mucinous cribriform pattern. PD-L1 staining was more heterogenous and weak due to abundant mucinous components.”

Therefore we showed the representative IHC images of ALK-positive lung cancer.

We have changed and added the higher magnification of the representative images for PD-L1 negative cases.

Since the authors claimed that EGFR wild type may be a predictor for PD-L1 expression (Figure 2), why did the authors only perform statistical analysis for the patients with EGFR mutations (Table 4)?

Response: We agree with the respectful suggestions given by the reviewer. In the NCCN guideline, immunotherapy is recommended as negative or unknown test results for EGFR mutations and ALK rearrangements. Therefore, there are limited and controversial data on PD-L1 expression in EGFR-mutated lung adenocarcinoma. Although EGFR wild type may be a predictor for PD-L1 expression, we also focused on the association with clinicopathological factors of PD-L1 expression in patients with EGFR mutated lung adenocarcinomas. Our results demonstrated that PD-L1 expression was significantly higher in patients with recognizing distinctive clinicopathological features, especially solid histologic pattern and higher stage in EGFR mutant group.

Minor comments:

The image quality of Figure 2 is low. What is the P value for the multivariate logistic regression analysis for individual variables?

Response: We have changed and added P-value in the Figure 2

The number of % for the PD-L1-positive neuroendocrine carcinoma in missing. line 88 A typo in line 93 “Supplementary Table 1”

Response: We have added missing data and word.

“Of 21 large cell neuroendocrine carcinomas, 6 (28.6%) were PD-L1 positive, including 5 (23.8%) with TPS of 1-49% and 1 (4.8%) with TPS of ≥ 50%.”

Line 116-line 118 is redundant. The same sentences showed in page 3, line 109-line 111.

Response: We have deleted line 116-118.

The titles of Tables 5, 6 and 7 are the same.

Response: We have changed the title of Table 7. However, the title of Table 5 and 6 are different as bellow:

Table 5: Multivariate logistic regression analysis of clinicopathological features for PD-L1 expression in lung adenocarcinoma (1% cutoff value for PD-L1 positivity).

Table 6: Multivariate logistic regression analysis of clinicopathological features for PD-L1 expression in lung adenocarcinoma (50% cutoff value for PD-L1 positivity).

 The Discussion is good.

Response: Thank you for your comment.

Reviewer 2 Report

Dear Authors

I read your work with great interest.

The work presented in the manuscript seems to have a high clinical significance due to the not fully explored the association of PD-L1 expression with EGFR and ALK mutations.The authors presented deep analyzes on IHC material and in conjunction with EGFR and ALK mutations. In addition, the advantage of the work is that it was performed on 1000 cases of lung cancer.

The only thing missing was the association of PD-L1 expression with EGFR and ALK mutations in the context of patient survival. However, due to the fact that the patients included in the study underwent resection in 2017-2019 it is understandable.

However, minor errors should be corrected:

Due to the fact that the work also concerns EGFR and ALK mutations, the introduction should contain the necessary information to understand why the authors decided to investigate their relationship with PD-L1.

There are no references to tables in the text.

Please pay attention not to repeat the results in the text that were included in the table. It is not necessary.

Figure 1 (b) is of poor quality. Please choose other pictures

In the methodology, the 4.3 EGFR and ALK should be described in separate sections. The authors used IHC as the main method, therefore it should also be extended with a visualization system. Was IHC made in any apparatus? The adding such information can make it easier to compare between different studies.

Please bold significant p in the tables, the reader more easily noticed.

In line 178-182, information on the exact number of individual subtypes should be in the methodology not in discussion.

Author Response

Reviewer(s)' Comments to Author: 

Reviewer 2

Dear Authors

I read your work with great interest.

The work presented in the manuscript seems to have a high clinical significance due to the not fully explored the association of PD-L1 expression with EGFR and ALK mutations. The authors presented deep analyzes on IHC material and in conjunction with EGFR and ALK mutations. In addition, the advantage of the work is that it was performed on 1000 cases of lung cancer.

The only thing missing was the association of PD-L1 expression with EGFR and ALK mutations in the context of patient survival. However, due to the fact that the patients included in the study underwent resection in 2017-2019 it is understandable.

However, minor errors should be corrected:

Due to the fact that the work also concerns EGFR and ALK mutations, the introduction should contain the necessary information to understand why the authors decided to investigate their relationship with PD-L1.

Response: As the reviewer rightly pointed out, we have added the necessary information to understand why the authors decided to investigate relationship with PD-L1 and genetic alteration in the introduction section of revised manuscript.

“Furthermore, immunotherapy is recommended as negative or unknown test results for EGFR mutations and ALK rearrangements in the NCCN guideline [1]. Therefore, there are limited and controversial data on PD-L1 expression in EGFR-mutated lung adenocarcinoma.”

There are no references to tables in the text.

Response: Thank you for your comment. We described the tables in the results (Table 1-6) and discussion (Table 7) section. Therefore, we have highlighted the tables in bold.

Please pay attention not to repeat the results in the text that were included in the table. It is not necessary.

Response: We have changed. Only data that we can highlight was rewritten.

Figure 1 (b) is of poor quality. Please choose other pictures

Response: We have changed the figure 1(b)

In the methodology, the 4.3 EGFR and ALK should be described in separate sections. The authors used IHC as the main method, therefore it should also be extended with a visualization system. Was IHC made in any apparatus? The adding such information can make it easier to compare between different studies.

Response: We have separated the section of EGFR and ALK in the materials and method. And we have added detailed IHC method in the materials and method.

4.3. Analysis of EGFR mutation

EGFR gene alteration was detected by either real-time PCR with PNA-clamping methods, direct sequencing, or both methods. The PNA-ClampTM EGFR mutation detection kit (PANAGENE, Inc., Daejeon, Korea) was used for realtime PCR, performed as described previously [39]. When detection was performed only with direct sequencing, exon 18, 19, 20, and 21 were sequenced as previously described [40]. When both methods were used, exons containing mutations detected by real-time PCR were sequenced, and exon 19 was sequenced if no mutation was detected by real-time PCR.

4.4. Analysis of ALK fusion

For the ALK fusion, ALK immunohistochemistry was performed using an anti-ALK mouse monoclonal antibody (clone: 5A4, Leica Biosystems Newcastle Ltd, UK; diluted 1:50) and the Leica Bond III automated system (Leica Biosystems Melbourne Pty Ltd) in 785 lung adenocarcinomas. The sections were incubated at pH 9 for 30min at 100°C. The ALK fusions in the 33 ALK-immunohistochemically positive cases were confirmed by fluorescence in situ hybridization (FISH). ALK FISH testing was performed using the Vysis ALK BreakApart probe kit (Abbott Molecular, Des Plaines, IL, USA) and a positive FISH result for ALK rearrangement was defined as >15% of tumor cells with a split signal.

“For the 22C3 pharmDx assay, sections were stained with anti-PD-L1 22C3 mouse monoclonal primary antibody using the EnVision FLEX visualization system on a Dako Autostainer Link 48 system with negative reagent controls and cell line run controls, as described in the PD-L1 IHC 22C3 pharmDx package insert [37]. Deparaffinization, rehydration, and target retrieval was performed with a 3-in-1 procedure using PT Link. Following peroxidase blocking, specimens were incubated with monoclonal mouse primary antibody to PD-L1 or the negative control reagent. Specimens were then incubated with Mouse Linker, followed by incubation with a ready-to-use visualization reagent consisting of secondary antibody molecules and horseradish peroxidase molecules coupled to a dextran polymer backbone. The enzymatic conversion of the subsequently added chromogen results in the precipitation of a visible reaction product at the site of the antigen. The color of the chromogenic reaction is modified using a chromogen enhancement reagent; the specimen may then be counterstained and cover slipped. Results were interpreted using a light microscope [38].”

Please bold significant p in the tables, the reader more easily noticed.

Response: We have addressed the significant P-value using bold P.

In line 178-182, information on the exact number of individual subtypes should be in the methodology not in discussion.

Response: Thank you for your comment. In discussion section, we mentioned it again to summarize and emphasize. As the reviewer rightly pointed out, we added this comment in materials and methods section of revised manuscript.  

“This study included 1000 patient samples including 785 adenocarcinomas, 188 squamous cell carcinomas, 21 large cell neuroendocrine carcinomas, 4 carcinoid tumor and 2 small cell carcinomas who underwent lung resection for lung mass between July 2017 and March 2019 at Samsung Medical Center.”

Round 2

Reviewer 1 Report

The revised manuscript has been improved appropriately.